# The Neuropeptide Cortistatin Alleviates Neuropathic Pain in Experimental Models of Peripheral Nerve Injury

**DOI:** 10.3390/pharmaceutics13070947

**Published:** 2021-06-24

**Authors:** Clara P. Falo, Raquel Benitez, Marta Caro, Maria Morell, Irene Forte-Lago, Pedro Hernandez-Cortes, Clara Sanchez-Gonzalez, Francisco O’Valle, Mario Delgado, Elena Gonzalez-Rey

**Affiliations:** 1Institute of Parasitology and Biomedicine Lopez-Neyra, IPBLN-CSIC, Parque Tecnologico de la Salud, 18016 Granada, Spain; claricapf@gmail.com (C.P.F.); rbenitez@ipb.csic.es (R.B.); martacm@ipb.csic.es (M.C.); maria.morell@genyo.es (M.M.); irene.forte@ipb.csic.es (I.F.-L.); csangon@ipb.csic.es (C.S.-G.); 2Genyo Center for Genomics and Oncological Research, Parque Tecnologico de la Salud, 18016 Granada, Spain; 3Department of Orthopedic Surgery, San Cecilio University Hospital, 18071 Granada, Spain; hdezcp@hotmail.com; 4Department of Pathology, School of Medicine, IBIMER and IBS-Granada, Granada University, 18016 Granada, Spain; fovalle2013@gmail.com

**Keywords:** neuropathic pain, neuropeptide, neuroprotection, neuroregeneration, peripheral nerve injury, neuroinflammation

## Abstract

Neuropathic pain is one of the most severe forms of chronic pain caused by the direct injury of the somatosensory system. The current drugs for treating neuropathies have limited efficacies or show important side effects, and the development of analgesics with novel modes of action is critical. The identification of endogenous anti-nociceptive factors has emerged as an attractive strategy for designing new pharmacological approaches to treat neuropathic pain. Cortistatin is a neuropeptide with potent anti-inflammatory activity, recently identified as a natural analgesic peptide in several models of pain evoked by inflammatory conditions. Here, we investigated the potential analgesic effect of cortistatin in neuropathic pain using a variety of experimental models of peripheral nerve injury caused by chronic constriction or partial transection of the sciatic nerve or by diabetic neuropathy. We found that the peripheral and central injection of cortistatin ameliorated hyperalgesia and allodynia, two of the dominant clinical manifestations of chronic neuropathic pain. Cortistatin-induced analgesia was multitargeted, as it regulated the nerve damage-induced hypersensitization of primary nociceptors, inhibited neuroinflammatory responses, and enhanced the production of neurotrophic factors both at the peripheral and central levels. We also demonstrated the neuroregenerative/protective capacity of cortistatin in a model of severe peripheral nerve transection. Interestingly, the nociceptive system responded to nerve injury by secreting cortistatin, and a deficiency in cortistatin exacerbated the neuropathic pain responses and peripheral nerve dysfunction. Therefore, cortistatin-based therapies emerge as attractive alternatives for treating chronic neuropathic pain of different etiologies.

## 1. Introduction

Clinical neuropathic pain represents a frequent and serious global public health issue that affects between 7 and 10% of the population worldwide [1], and this percentage is expected to rise in the next decades [2]. It is estimated that around 40% of patients attending pain clinics for treatment exhibit neuropathic symptoms. Neuropathic pain is a consequence of nerve injury caused by direct trauma of the nerve, tumor growth or therapy, nerve compression, or autoimmune and metabolic disorders such as diabetes [3,4]. Hypersensitivity (manifesting as spontaneous pain), allodynia (pain in response to normally innocuous tactile or thermal stimuli), and hyperalgesia (an exaggerated response to subsequent noxious stimuli as thermal or mechanical pressure) are the dominant features of chronic, pathological neuropathic pain. Despite tremendous efforts being focused on both clinical and preclinical research, the management of neuropathic pain remains an unmet clinical need [5].

Treatment of pain-related suffering requires knowledge of how pain signals are initially interpreted and subsequently transmitted and perpetuated. Persistent neuropathic pain results from multiple changes in the peripheral and central nervous systems, including increased excitability and reduced thresholds of primary sensory neurons, altered spinal cord synaptic processing, loss of inhibitory interneurons, and modifications of the brainstem input to the spinal cord [3]. Among these, changes in primary nociceptive neurons are critical in sensing, initiating, and perpetuating neuropathic pain, as they give rise to C- and Aδ-axons that can be activated by noxious mechanical, thermal, or chemical stimuli and to Aβ-axons that can be activated by nonnoxious tactile stimuli. The inflammation is a critical component of the progression and maintenance of the pathophysiology of neuropathic pain [6], in which the Wallerian degeneration of axons and myelin sheaths occurring after nerve injury is accompanied by the secretion of inflammatory cytokines, chemokines, and oxidative mediators, the recruitment of macrophages and other immune cells, and the activation of resident astrocytes and microglial cells [7]. The components of this inflammatory milieu are responsible of evoking allodynic and hyperalgesic responses in part because of a persistent sensitization of the nociceptive neurons and partially by the initiation of a facilitated state of processing of the afferent input in the spinal cord.

Because the balance between excitation and inhibition is crucial for maintaining normal sensory function, inhibitory signals must participate in pain perception, limiting the duration and intensity of pain. Endogenous opioids and cannabinoids are well-known analgesic factors and GABAergic inhibitory interneurons play a critical role in the modulation of pain processing at the spinal level [8,9]. Moreover, nociceptive neurons have the plasticity to produce antinociceptive peptides (i.e., neuropeptide Y, galanin, and enkephalins) in response to injury and inflammation [10]. Therefore, the identification of new endogenous analgesic peptides is critical to understand the process of nociception and to develop novel strategies for the treatment of chronic pathological pain.

Cortistatin is a cyclic neuropeptide discovered in the brain cortex and hippocampus based on its capacity to suppress neuronal activity [11]. Due to its high homology with somatostatin, cortistatin is able to bind to all somatostatin receptors (sstr1–5). However, the literature indicates that cortistatin exerts various effects on the nervous system and peripheral tissues that are not shared with somatostatin [12], probably due to the capacity of cortistatin, but not somatostatin, to bind to receptors other than sstr, including ghrelin receptor (GHSR1), Mas gene-related receptor X-2, and a still-unidentified cortistatin-selective receptor [13,14]. Cortistatin has also been identified as a potent anti-inflammatory agent that protects from the development of a variety of inflammatory and autoimmune disorders [15] and as a critical regulator of vascular function [16,17,18]. Moreover, various studies strongly suggest that cortistatin is a critical analgesic factor that regulates inflammatory-induced pain in many preclinical conditions [19,20,21,22]. The evidence indicates that cortistatin induces the relief of inflammatory pain through peripheral and central mechanisms that are independent of its anti-inflammatory activity and that involve the blockade of intracellular signals that drive pain transmission and neuronal plasticity in primary nociceptive neurons and the regulation of pain-induced sensitization of secondary neurons in the spinal cord [21,22]. However, despite the well-known analgesic effect in persistent inflammatory pain, the role played by cortistatin in neuropathic pain is unknown. Therefore, the aim of this study was to evaluate the potential analgesic action of cortistatin in different established preclinical models of neuropathic pain induced by peripheral nerve injury as a consequence of chronic constriction of the nerve, partial or complete nerve transection (with or without reconstruction), or severe diabetes [23]. We will also investigate how a deficiency in this neuropeptide could affect the sensory hypersensitivity developed during these conditions, as well as the potential cellular mechanisms and receptors that could be involved in its anti-nociceptive effects.

## 2. Materials and Methods

### 2.1. Animals and Ethic Statement

The experiments reported in this study followed the ethical guidelines for investigations of experimental animals approved by the Animal Care and Use board and the Ethical Committee of Spanish Council of Scientific Research (Animal Care Unit Committee IPBLN-CSIC # protocol 202-11-3), and they were performed in accordance with the guidelines from Directive 2010/63/EU of the European Parliament on the protection of animals used for scientific purposes and for investigations of experimental pain in conscious animals. Animal studies are reported in compliance with the ARRIVE v.2.0 guidelines [24]. Mice lacking the gene for cortistatin (CST-KO) were a generous gift of Dr. Luis de Lecea (Stanford University, La Jolla, CA, USA) and were generated in a C57BL/6 background and backcrossed with C57BL/6 mice for ten generations, as previously described [25]. Mice heterozygous (CST-het) for cortistatin were generated by crossing female CST-KO and male wild-type (CST-wt) mice. CST-het breeding pairs were used to generate a littermate colony of CST-wt, CST-het, and CST-KO mice for cortistatin. Both male and female mice (20–24 g body weight, 8–12 weeks-old) were used in all experiments described in this study, and no sex differences were found. All animals were housed in groups of 10 mice per cage in a controlled-temperature/humidity environment (22 ± 1 °C, 60–70% relative humidity, with wood shaving bedding and nesting material), with a 12 h light/dark cycle (lights on at 7:00 a.m.) and fed with rodent chow (Global Diet 2018, Harlan) and tap water ad libitum. Mice were allowed to acclimatize to the experimental room for one hour before experiments. Mice were randomly assigned to the different experimental groups. Experiments were designed to make sample sizes relatively equal. Power calculations were performed using the software G*Power (www.gpower.hhu.de, accessed on 24 July 2020) to ensure that adequate group sizes were used for the studies detailed below. We calculated a minimum size of five to eight mice per group in order to have a power > 0.95 of detecting approximately a 30% change, assuming a standard deviation of 30% at a significance level of *p* < 0.05, expecting an effect size of 1.82 for ANOVA tests. The Ethical Committee established as humanitarian end points the observation of a sustained body weight loss higher than 15% for two days (especially important for mice subjected to the diabetes model), impossibility of the animal to access food and water (even facilitated in the cage bed), evident signs of pain (assessed by maintained audible groans), and/or signs of limb mutilations. With the exception of one animal that suffered self-mutilations in the affected hind-paw during the sciatic nerve transection model and was sacrificed, none of the mice reached the described humanitarian end points along the study.

### 2.2. Reagents

Mouse cortistatin-29, mouse acylated-ghrelin, and somatostatin-28 were purchased from Bachem (Bubendorf, Switzerland), and streptozotocin (STZ), cycloSOM, GHRP6, CYN-154806, formalin, octreotide, naloxone, and pertussis toxin (PTX) from Sigma-Aldrich (St. Louis, MO, USA). BIM-28163 and BIM-23867 were generously provided by Dr. M.D. Culler (Ipsen). All drugs were dissolved in physiological saline (0.9% NaCl), except naloxone that was dissolved in 1% ethanol. We previously found that 1% ethanol did not alter nocifensive responses when administered centrally.

### 2.3. Induction and Treatment of Acute Pain Evoked with Formalin

We evoked acute inflammatory pain by the intraplantar (i.pl.) injection of formalin (5%, 20 µL) in the hind paw and assessed the biphasic spontaneous nocifensive responses by measuring the time the mice spent on licking or flinching the affected paw for 45 min. Mice received cortistatin, ghrelin, and somatostatin 15 min before formalin through three routes: peripherally by i.pl. injection in the plantar surface of the hind paw at 100 ng in 20 µL of saline, centrally by intrathecal (i.t.) injection in the lumbar region (between L5 and L6 level) at 10 ng in 10 µL of saline, and systemically by intraperitoneal (i.p.) injection at 1 µg in 200 µL of saline. These doses were used in the base of our previous experience with other pain models [21,22]. The mice received saline (same volume and injection pathway as described for peptides) as a vehicle control. To study the involvement of specific receptors, mice received cortistatin-receptor antagonists (BIM-28163, BIM-23867, CYN-154806, cycloSOM, and GHRP6) centrally (i.t., 5 µM, 10 µL) 1 h before cortistatin. The use of these antagonists was based on previous data from various reported in vivo studies [26,27,28,29,30]. To study the involvement of Gαi-coupled receptors, pertussis toxin was given in two i.t. injections (2 × 200 ng) 24 and 12 h prior to formalin injection. To study the involvement of opioid receptors, mice received i.t. naloxone (2 µg, in 1% ethanol) 30 min before cortistatin.

### 2.4. Induction and Treatment of Chronic Constriction Injury (CCI) of Sciatic Nerve

To induce neuropathic pain by the unilateral CCI of the sciatic nerve, mice were anesthetized (i.p., ketamine 80 mg per kg mouse, Richter Pharma; xylazine 10 mg per kg mouse, Fatro Iberica), an incision was made on the shaved right lateral thigh, and the sciatic nerve was exposed between the vastus lateralis and the biceps femoris. Three ligatures (6-0 silk) were then placed around the sciatic nerve proximal to the trifurcation with 1 mm spacing between them. We tied the ligatures until they just slightly constricted the diameter of the nerve, and we observed a brief twitch in the respective hind limb. This corresponded to blanching of the nerve and a significant reduction in nerve blood flow secondary to occlusion of the epineural vasculature. The muscle incision and skin were closed in two layers with 6-0 silk sutures. We used a control group of sham-operated mice, in which the sciatic nerve was exposed but no ligature was applied. After surgery, all animals received 0.5 mg/kg of atipemazole (Braun) i.p. as an antagonist for the anesthesia to hasten the recovery, and they were kept in a warming blanket for several hours. We also used subcutaneous (s.c.) analgesics butorphanol (2 mg/kg; Richter Pharma) and meloxicam (2 mg/kg; Norbrook) for 72 h post-surgery. Treatment consisted of the injection of cortistatin at the local level (1 µg in 20 µL of saline, s.c. around the surgery site) or at the systemic level (2 µg in 200 µL of saline, i.p.), every other day, starting two days after CCI until day 14. Mice received saline (same volume and injection pathway, as described for cortistatin) as a vehicle control. To study the analgesic effect of cortistatin in the established CCI, mice were treated with a single i.t. (20 ng in 10 µL of saline) or s.c. (1 µg in 20 µL of saline) injection of cortistatin five days after CCI surgery. When indicated, mice received cortistatin-receptor antagonists (cycloSOM and GHRP6) locally (s.c., 0.5 mM, 20 µL) or centrally (i.t., 5 µM, 10 µL) one hour before cortistatin, or naloxone (2 µg, i.t.) 30 min before cortistatin. We assessed thermal hyperalgesia, mechanical hypersensitivity, and mechanical allodynia in the hind paws (ipsilateral and contralateral to the injured nerve) at different times after CCI, as described below. Five days after CCI, we determined the contents of inflammatory mediators, neurotrophic factors, and cortistatin in the sciatic nerve and lumbar dorsal spinal cord segments (ipsilateral to the side of CCI), as described below.

### 2.5. Induction and Treatment of Spared Nerve Injury (SNI)

To induce the SNI model, mice were anesthetized with i.p. ketamine-xylazine, and the sciatic nerve was exposed at the mid-thigh level, as described above for CCI. The three terminal branches of the sciatic nerve (tibial, common peroneal, and sural nerves) were carefully separated while minimizing any contact with or stretching of the sural nerve. The tibial and common peroneal nerves were then individually ligated with 6.0 silk, cut distally, and 2–3 mm of each nerve distal to the ligation was removed. The muscle and skin incisions were closed in two layers with silk sutures. Sham-operated mice, in which the sciatic nerve terminal branches were exposed but neither ligated nor transected, were used as the controls of reference. Treatment consisted of the injection of cortistatin at the local level (1 µg in 20 µL of saline, s.c. around the surgery site), every other day, from days 2 to 15 after SNI-surgery. Mice received saline (20 µL, s.c.) as a vehicle control. To study the analgesic effect of cortistatin in established SNI, mice were treated with a single i.t. (20 ng in 10 µL of saline) injection of cortistatin five days after SNI surgery. We assessed thermal hyperalgesia and mechanical and cold allodynia in the hind paws (ipsilateral to surgery) at different times after SNI, as described below.

### 2.6. Induction and Treatment for the Sciatic Nerve Transection (SNT) Model

To test the role of cortistatin in peripheral nerve regeneration after injury, we used a SNT model (Appendix A). Mice were anesthetized by i.p. ketamine-xylacine injection and the right sciatic nerve was exposed at the mid-thigh level, as described above for CCI. A biocompatible resorbable bilayer collagen membrane (4 × 4 mm; Geistlich Bio-Gide, Wolhusen, Switzerland) was then collocated under the nerve, which was fixed to the membrane surface with a 10-0 Syneture nylon monofilament (Covidien, Dublin, Ireland) to avoid torsional mismatching after sectioning. After that, the sciatic nerve was sectioned at 1 mm from the membrane sutures, and a segment of 2 mm-length in the distal portion of the nerve was removed. We wrapped the two nerve stumps with the membrane, forming a tubular conduit to allow axons guidance [31], and closed the free edges with 7-0 polypropylene sutures (Covidien, Dublin, Ireland) The muscle and skin wound were closed, and anesthesia recovering and administration of the post-operative analgesic were performed, as described above for CCI. Cortistatin was administered locally (3 µg in 50 µL of saline s.c, around the surgery site) for six weeks, every other day (three times per week) starting three days after nerve transection. Saline instead of cortistatin was used as a vehicle control. Twenty-four hours previous to the surgery and at different points after transection, each mouse was video recorded during walking for 60–90 s to evaluate the clinical score and to quantify the lateral foot-base angle. Thermal and tactile neuropathic pain responses were measured in the hind paws contralateral (unlesioned) and ipsilateral (lesioned) to the injured nerve four weeks after transection. After sacrifice (6 weeks after surgery), the gastrocnemius muscles were dissected and collected from both operated and contralateral limbs, fixed in 10% buffered-formalin, and weighted. The percentage of gastrocnemius muscle mass reduction was determined as follows: [(contralateral muscle weight—ipsilateral muscle weight)/contralateral muscle weight] × 100. After this, the gastrocnemius muscles were processed for histopathological analysis, as described below. Moreover, the sciatic nerve–thigh muscle complexes of both limbs were dissected and subsequently formalin-fixed for further immunohistochemistry determination, as described below.

### 2.7. Induction and Treatment of Neuropathic Pain in Diabetic Mice

To induce diabetes, mice were injected i.p. with a single high-dose of streptozotocin (STZ, 200 mg/kg, freshly prepared in citrate buffer 100 mM, pH 4.5). The tail vein blood glucose levels were measured weekly for five weeks after STZ injection by using a glucometer and electronic sticks from Bayer (always measured at 10:00 a.m., mice were not previously fasted). Mice showing a glucose level above 400 mg/dL two weeks after STZ injection were considered diabetic and selected for further treatments. Mice injected i.p. with citrate buffer were used as nondiabetic controls. Treatment consisted of the injection of cortistatin at the systemic level (2 µg in 200 µL of saline, i.p.), every other day, from days 14 to 28 after STZ injection. Mice received saline (200 µL, i.p.) as a vehicle control. To study the analgesic short-term effect of cortistatin in established diabetes, mice were treated with a single injection of cortistatin through central (i.t., 20 ng in 10 µL of saline), local (i.pl., 1 µg in 10 µL of saline), or systemic (i.p., 2 µg in 200 µL of saline) pathways on day 17 after STZ injection. We assessed thermal hyperalgesia and mechanical allodynia in the hind paws at different times after diabetes induction or after cortistatin injection, as described below. Four weeks after STZ injection, sciatic nerves were dissected and collected, and the contents of inflammatory mediators in nerves were determined as described below.

### 2.8. Analysis of Functional Tests in the SNT Model

Dysfunction of the sciatic nerve after its transection was evaluated using a functional clinical score and by measuring the lateral foot-base angle (LFBA) as follows.

-Determination of the clinical score: to assess the functional nerve recovery following SNT, walk tracking videos were recorded (Sony HDR-XR260VE) for each mouse one day before and at different times (7, 17, and 42 days) after nerve lesion, and they were evaluated by two independent blinded examiners to obtain four parameters affecting locomotion: gait, toe spreading, ankle dorsiflexion, and foot step [32,33]. Each of these parameters were individually scored on a 0–2 scale as follows, giving a maximum possible score of 8 per animal: Gait: 0—no motor deficits, 1—mild loss of coordination (manifests in an uneven gait or waddling), 2—walking deficits characterized by lateral disequilibrium (weak support with the lesion); toe spreading: 0—normal toes separation, 1—extended toes close together, 2—curved toes tightly close; ankle dorsiflexion (based on the lateral foot-base angle measurements, see below): 0—ankle angle > 90°, 1—ankle angle of 45–90°, 2—ankle angle < 45°; foot step: 0—normal foot dorsiflexion in the steps, 1—partial loss of foot dorsiflexion, 2—total absence of foot dorsiflexion and rigidity (ipsilateral steps are characterized by “flat” foot).-Measurement of the LFBA: left- and right-side views of one walking trial for all mice were captured once one day before and at different time-points (7, 17, and 42 days) after surgery. The video sequences were examined using VLC 3.0.12 multimedia player platform (available at https://www.videolan.org/vlc/, accessed on 24 July 2020), in order to select adequate frames, in which the animals could be seen in defined phases of the step cycle. The LFBA was measured using ImageJ software, as previously described [33]. Briefly, side-view frames showing the ipsilateral foot at the toe-off position with maximal plantar flexion and the contralateral foot in the initial stance phase were evaluated, in which the angle is formed by a line parallel to the mid- and hind foot (i.e., excluding the toes) and the horizontal line. Four to six frames were evaluated for each mouse and the mean of these values was considered as LFBA for each time-point.

### 2.9. Measurement of Nocifensive Responses to Peripheral Nerve Injury

All the nociceptive behavior assays were performed in a blinded manner by an independent observer.

-Thermal nociceptive responses were determined using Hargreaves’s radiant heat apparatus (IITC Life Sciences). Briefly, mice were placed in Plexiglas boxes placed on a glass floor maintained at 30 °C and allowed to adjust for at least one hour. After acclimation, mice received an automatic heat source (50 W/10 V) onto the plantar surface of the hind paw and the paw withdrawal latency (PWL) was recorded. Each mouse was tested three times (with one-minute intervals between consecutive measurements) and the latencies were averaged for each animal. We adjusted the basal PWL to 9–12 s and set a cut-off of 20 s (defined as complete analgesia) to prevent tissue damage.-Mechanical pressure hypersensitivity in the hind paw was determined by using a Randall–Selitto Pressure Analgesiometer (IITC Life Sciences). Briefly, mice were restrained as such where they were able to flex their legs. A constantly increasing pressure was applied to the plantar surface of both the ipsilateral and contralateral hind paws through the tip of the paw pressure analgesiometer. Readings (in g) were obtained when the mice attempted to withdraw the paw, and the results were expressed as the decrease (in percentage) in threshold response to pressure stimulation between contralateral and ipsilateral hind paws. The cut-off force was set at 200 g to prevent further injury to the paws.-Mechanical/tactile allodynia was determined by quantifying the withdrawal threshold of the hind paw in response to stimulation with flexible von Frey filaments (range 0.02–3.0 g; IITC Life Sciences). Mice were placed in Plexiglas boxes on a stainless-steel mesh floor and allowed to habituate for at least 30 min. We then applied a series of calibrated von Frey hairs (six times each on the basis of the Dixon’s up-and-down method) perpendicularly to the lateral and medial plantar surface of the ipsilateral and contralateral hind paw with sufficient force to bend the filament for 4–5 s. Brisk withdrawal or paw flinching was considered positive responses. In the absence of a response, a filament with the next greatest force was applied. In the presence of a response, a filament with the next lowest force was applied. We recorded the 50% withdrawal threshold (i.e., force of the von Frey hair to which an animal reacts in 50% of the presentations).-Cold allodynia was determined via acetone test, as previously described [34]. A drop of acetone solution was delicately dropped onto the lateral plantar surface of the paw, using a blunt needle connected to a syringe without touching the skin. Acetone was applied three times to the ipsilateral hind paw at intervals of 30 s, and the duration of biting or licking of the hind paw was recorded. We reported the cumulative time of biting/licking in all three measurements with an arbitrary minimal value of 0.5 s and a maximum of 10 s in each of the three trials.

### 2.10. Histomorphometric Analysis

To analyze the histopathological signs in the SNT model, the gastrocnemius muscle, as well as the sciatic nerves and the surrounding soft tissues of both body sides (ipsilateral and contralateral to the operated limb), were collected, fixed in 10% buffered-formalin, and paraffin-embedded. The gastrocnemius samples were cross-sectioned (4 μm-thick) and stained with hematoxylin-eosin following standard techniques. Images were randomly acquired on a millimeter scale in the eyepiece of an Olympus BH2 microscope (objective 40×), and the number of muscle fibers per area was quantified. On the other hand, the proximal and distal segments of the sciatic nerve–muscle complex with respect to the transection site were separately cross-sectioned (4 μm) and processed for immunohistochemical analysis of myelin basic protein (MPB)-positive myelinated axons. Briefly, tissue sections were heat-treated (95 °C, 20 min) in 1 mM of EDTA pH 8.0 buffer for antigenic unmasking in a PT module (Thermo Fisher Scientific, Kalamazoo, MI, USA), endogenous peroxidase-blocked with peroxidase solution (10 min, room temperature), incubated (30 min, room temperature) with a prediluted ready-to-use rabbit anti-MBP polyclonal (Vitro-Master Diagnostica, Granada, Spain), and detected with the micropolymer-peroxidase-based method (Ultravision Quanto from Thermo Fisher Scientific, Waltham, MA, USA) and diaminobenzidine (Vitro-Master Diagnostica, Granada, Spain), using an automatic immunostainer (Autostainer480, Thermo Fisher Scientific, Waltham, MA, USA). Samples were finally hematoxylin-counterstained. We used cross-sections of the contralateral sciatic–muscle complex (corresponding to similar locations to proximal and distal ipsilateral segments) as internal controls (unlesioned) for each mouse. Data were gathered on the presence of MBP-labeled fibers that were quantified in the proximal and distal segments on a millimeter scale in the eyepiece of an Olympus BH2 microscope (objective 40×).

### 2.11. Measurement of Neuropeptide, Neurotrophin, and Cytokine Contents

To determine the levels of neuropeptide, neurotrophic factors, and cytokines in tissues of the nociceptive system, the dorsal root ganglia (DRGs) and dorsal horn spinal cord halves of the lumbar region (L4-L5 segment) and sciatic nerves, ipsilateral and contralateral to the surgery or formalin injection side, were separately dissected and isolated. At least two tissue samples (50 mg) from the same group were pooled and homogenized in 1 mL of lysis buffer (50 mM of Tris-HCl, pH 7.4, with 0.5 mM of DTT, and 10 μg/mL of a cocktail of proteinase inhibitors containing phenylmethylsulfonyl fluoride, pepstatin, and leupeptin, all from Sigma-Aldrich). Samples were centrifuged (20,000× *g*, for 15 min, at 4 °C) and the supernatants were stored at −80 °C until their use. The concentration of proteins in extracts was assessed using a bicinchoninic acid BCA Protein Assay Kit (Pierce/Thermo Fisher). The contents of neuropeptides, cytokines, and neurotrophic factors in the protein extracts were determined with specific sandwich or competitive ELISAs according to the manufacturers’ recommendations: cortistatin and CGRP from Phoenix Pharmaceuticals, glial cell line-derived neurotrophic factor (GDNF) from Abcam, brain-derived neurotrophic factor (BDNF) from R&D Systems, TNFα and IL-6 from BD Pharmingen, and IL-1β and CCL2/MCP1 from PeproTech. According to the companies’ information, the capture and biotinylated antibodies and ELISA kits were specific for the corresponding neuropeptides, neurotrophins, and cytokines and did not cross-react with other related peptides and proteins.

### 2.12. Determination of Gene Expression

We assessed the gene expression of CGRP, Substance P (SP), inducible nitric oxide synthase (iNOS), and cycloxygenase-2 (COX2) in RNA isolated from DRGs and sciatic nerves by real-time qRT-PCR, as previously described [22,35]. In brief, total RNA was isolated from tissue samples by homogenization in EZNA HP total RNA Kit-Animal Tissue (Omega Bio-Tek) following the manufacturers’ protocols. Precipitated RNA (1 µg) was treated with DNase I (1 U) and then reverse-transcribed using the RevertAid First Strand cDNA Synthesis Kit (200 U, Thermo Fisher, Waltham, MA, USA) and random hexamer primers (5 µM) at 42 °C for 60 min in a Mastercycler EP Gradient Thermocycler (Eppendorf). SYBER green quantitative PCR (SensiFast Sybr No-Rox mix, from Bioline, Memphis, Tennessee, USA) was performed on the CFX96 Real-Time PCR system (Bio-Rad) using the following conditions: 94 °C for five minutes followed by 40 cycles at 94 °C for 30 s, annealing at 60 °C for 30 s, and extension at 72 °C for 30 s. The sequence of primer sets used were (5′ to 3′): SP FW: TCTTTTTTCTCGTTTCCACTCAA; SP RV: CATTAATCCAAAGAACTGCTGA’; CGRP FW: CTCCAGGCAGTGCCTTTGA; CGRP RV: CAGGTGGCAGTGTTGCAGG; iNOS FW: GTTCTCAGCCCAACAATACAAGA; iNOS RV: GTGGACGGGTCGATGTCAC; COX2 FW: AACCGCATTGCCTCTGAAT; COX2 RV: CATGTTCCAGGAGGATGGAG. We employed RNA 18S for normalizing the cDNA content in the samples (18S FW: 5′-CCCATTCGAACGTCTGCCCTATC-3′; 18S RV: 5′-TGCTGCCTTCCTTGGATGTGGTA-3′). Fold change was estimated with the ΔΔCt method.

### 2.13. Statistical Analysis

All data are expressed as mean ± SEM. We analyzed the data for statistical differences between two groups using the unpaired Student’s *t*-test or the nonparametric Mann–Whitney U-test. For multiple group comparisons, we used regular one-way ANOVA and post-hoc Bonferroni’s tests, or the nonparametric equivalence Kruskal–Wallis analysis of variance test. All analyses were performed using GraphPad Prism v8.0 software (La Joya, CA, USA). We considered *p*-values < 0.05 (two-tailed) as significant. The experiments were performed and analyzed in a randomized and blinded fashion. No data were excluded from the analysis.

## 3. Results

### 3.1. Cortistatin Alleviates Acute Pain Evoked by Formalin

First, we confirmed the analgesic effect of cortistatin in an acute inflammatory pain using a mouse model induced by i.pl. injection of formalin. Formalin induces two phases of spontaneous pain behavior. The initial neurogenic phase is caused by direct activation of nociceptors via Aδ and C-fibers, whereas the second tonic phase appears to be dependent on the combination of an inflammatory reaction in the peripheral tissue and functional changes in neurons of the dorsal horn spinal cord, which become sensitized to nociceptive stimuli [36]. In order to discriminate the site of action for cortistatin, we used three routes of cortistatin administration: systemic (i.p.), peripheral (i.pl.), and central (i.t. by lumbar puncture). We found that pre-emptive injection of cortistatin by any of these routes decreased first-phase neurogenic pain and second-phase inflammatory pain behavior, the spinal delivery being the most effective (Figure 1a). These results suggest both peripheral and central mechanisms of action for cortistatin in the attenuation of nociception. Notably, cortistatin showed improved analgesic effects than the related neuropeptides somatostatin and ghrelin, and the stable sstr2/3/4-agonist octreotide in inflammatory pain (Figure 1b). Among them, only cortistatin and ghrelin alleviated the neurogenic phase (Figure 1b).

We next investigated the involvement of specific receptors in the cortistatin’s analgesic action in formalin-induced acute pain. As cortistatin binds to somatostatin-receptors (sstr1–5), and ghrelin-receptor (GHSR1) and some of these receptors, including sstr1, sstr2, and GHSR1, are differentially expressed in murine DRG and spinal cord neurons [30,37,38,39], they emerge as potential mediators of cortistatin’s analgesic effects. Indeed, central injection of a nonselective sstr-antagonist (cyclosomatostatin) or two specific GHSR1-antagonists [29] partially reversed cortistatin-induced analgesia on the second-phase of formalin-induced pain (Figure 1c). Co-injection of sstr- and GHSR1-antagonists at the spinal level showed an additive effect and almost completely abrogated the analgesia caused by cortistatin in formalin-induced inflammatory pain (Figure 1c). Furthermore, the selective sstr2-antagonist CYN-154806 [26,38] reversed partially the cortistatin actions in inflammatory pain (Figure 1c). However, we observed no reversal effect by the antagonist for sstr5 BIM-23867 (Figure 1c), as sstr5 is not expressed in the murine nociceptive system [30]. Interestingly, the spinal injection of pertussis toxin, an inhibitor of the G protein subunit Gαi, abolished cortistatin actions in second-phase inflammatory pain (Figure 1d), supporting the involvement of Gαi-protein-coupled receptors, such as sstr2 and GHSR1 [40,41], and the inhibition of cAMP/PKA and channel-dependent [Ca^2+^]i, which are generally related to nociceptive signaling [8]. Interestingly, neither sstr nor GHSR1 participated in the cortistatin action in acute neurogenic pain (Figure 1c). However, the opioid-receptor antagonist naloxone prevented its analgesic action in formalin-induced first-phase neurogenic pain (Figure 1d). As expected, central and peripheral injection of cortistatin impaired the formalin-induced expression of the pronociceptive peptides substance P and CGRP in primary DRG neurons and their release by central terminals at the dorsal spinal level (Figure 1e). These findings confirmed that cortistatin alleviates neurogenic and inflammatory persistent pain at both peripheral and central levels.

### 3.2. Cortistatin Attenuates Neuropathic Pain Caused by Peripheral Nerve Injury

We next investigated the capacity of cortistatin to induce neuropathic pain relief in various experimental mouse models of sciatic nerve injury caused by chronic constriction of the nerve (CCI), by partial transection of the nerve (SNI), or by a metabolic disorder such as diabetes.

As expected, CCI of the sciatic nerve led to mechanical/tactile allodynia (von Frey test), heat hyperalgesia (Hargreaves test), and pressure hypersensitivity (Randall–Selitto test) (Figure 2a), similar to those seen in painful human peripheral neuropathies [42].

Repetitive systemic or local injections of cortistatin around the injured nerve during two weeks significantly reduced all the sensory signs of hyperalgesia and allodynia caused by CCI (Figure 2a). The anti-allodynic and hyperalgesic effect of cortistatin was rapidly observed after a short period of treatment (Figure 2a). Pre-emptive treatment with a GHSR1-antagonist almost fully abolished the sustained analgesic actions of cortistatin in CCI-induced neuropathic pain, and the blockade of somatostatin- and opioid-receptors partially reversed them (Figure 2b), suggesting the involvement of these receptors. Interestingly, a single i.t. lumbar injection of cortistatin significantly attenuated thermal and tactile hyperalgesia in mice with fully developed nerve injury, whereas its injection at the peripheral level only transitorily mitigated the nocifensive responses to heat and mechanical stimulation of the ipsilateral hind paw (Figure 2c). We observed that the anti-allodynic effects of cortistatin at the central level were abolished by blocking spinal GHSR1 and opioid-receptors, but not sstr (Figure 2c).

We next evaluated the effect of cortistatin in a mouse model of SNI [43], in which transection of the tibial and common peroneal branches of the sciatic nerve led to persistent neuropathic pain, manifested by significant thermal hyperalgesia, as well as allodynia in response to mechanical (von Frey test) and cold (acetone-drop test) stimulation in the territory of the intact sural branch (Figure 3a).

Two weeks of treatment with cortistatin around the injured nerves resulted in a rapid and marked alleviation of these signs of hypersensitivity, a long-lasting effect that was maintained once the treatment ceased (Figure 3a). Moreover, a single injection of cortistatin at the spinal level induced a significant relief in the mechanical allodynic responses in animals with established SNI-induced neuropathic pain (Figure 3b).

Finally, we examined the potential analgesic effect of cortistatin in the neuropathic pain manifestations that appears during the course of diabetes, using a recognized mouse model of streptozotocin (STZ)-induced diabetic polyneuropathy [44]. These animals develop primarily distal axon loss, systemic injury of the peripheral nerves, and altered axon–Schwann cell interactions. The injection of a high-dose of STZ, an alkylating antineoplastic agent that is particularly toxic for insulin-producing β-cells of the pancreas, induced persistent hyperglycaemia in mice, which correlated with the emergence of marked signs of hypersensitivity to thermal and mechanical stimuli in hind paws (Figure 4a).

The systemic treatment with cortistatin of animals with fully established diabetes progressively abolished heat hyperalgesia and mechanical allodynia, without correcting the blood glucose levels (Figure 4a). Interestingly, a single injection of cortistatin via peripheral and systemic pathways, but especially at the spinal level, transitorily alleviated diabetes-induced neuropathic pain (Figure 4b).

These findings suggest that cortistatin attenuates neuropathic pain caused by different types of sciatic nerve injury acting at peripheral and central levels through mechanisms that could involve control of both local/nerve responses and spinal sensitization.

### 3.3. Cortistatin Regulates Inflammatory and Neurotrophic Mediators of Neuropathic Pain

Because evidence indicates that the release of inflammatory cytokines and chemokines, as well as oxidative mediators in the injured nerve and spinal cord, play a critical role in the progression of nerve damage and pain manifestations, and that their balance with neurotrophic factors determines the capacity to regenerate or avoid the degeneration of the injured nerve [45,46,47,48,49,50], we next investigated whether treatment with cortistatin affected any of these mediators of neuropathic pain in two different preclinical scenarios: CCI of sciatic nerve and STZ-induced diabetic neuropathy. In the first place, we observed that CCI of the sciatic nerve resulted in significant increases in the levels of the inflammatory cytokines TNFα, IL-6, and IL-1β and of the chemokine CCL2 in the injured nerve and the ipsilateral dorsal horn of the spinal cord (Figure 5a).

Moreover, the CCI of sciatic nerve upregulated the local expression of the enzymes iNOS and COX2, which produce inflammatory and nerve excitatory factors, such as PGE2 and nitric oxide, that are involved in the development of neuropathic pain (Figure 5a). Treatment with cortistatin significantly reduced the levels of all these nociceptive mediators at both peripheral and spinal levels, and decreased the expression of CD68, a marker of infiltrating macrophages and activated microglia (Figure 5a). While it impaired the neuroinflammatory response, cortistatin corrected the levels of the neurotrophic factor GDNF, which is significantly decreased after CCI in both sciatic nerve and spinal cord (Figure 5a). In contrast, the levels of BDNF were differentially regulated by cortistatin during CCI, with the lack of effect on the CCI-induced upregulation of this neurotrophic factor in the sciatic nerve, but with a significant inhibition in its secretion at the spinal level (Figure 5a). Similar to that observed in the CCI model, systemic treatment with cortistatin partially avoided the increase in the levels of inflammatory cytokines and iNOS in the sciatic nerve of diabetic mice (Figure 5b). These findings suggest that cortistatin could ameliorate the persistent signs of neuropathic pain caused by peripheral nerve injury by impairing neuroinflammatory responses while promoting a neurotrophic protective/regenerative milieu at peripheral and central levels.

### 3.4. Cortistatin Plays an Endogenous Regulatory Role in the Development of Neuropathic Pain

Previous studies demonstrated that cortistatin is produced by the murine nociceptive system, especially in response to some inflammatory-mediated pronociceptive stimuli [21,22]. We found that two of the conditions used in this study to evoke nociceptive responses, such as formalin-induced acute pain (Figure 6a) and CCI-induced neuropathic pain (Figure 6b), increased the levels of cortistatin in the lumbar dorsal horn of the spinal cord, lumbar DRGs, and sciatic nerve, specifically at the ipsilateral side of the affected paw/nerve.

Therefore, we next examined the severity and duration of neuropathic painful responses to peripheral nerve injury in animals lacking cortistatin (CST-KO). We observed that a deficiency in cortistatin predisposed them to suffer exacerbated and longer nociceptive responses to tactile, thermal, and pressure stimulations in animals subjected to CCI of the sciatic nerve (Figure 7a–c). The increased mechanical and thermal hypersensitivity observed in CST-KO mice in comparison to CST-wt animals correlated with an increased imbalance between inflammatory cytokines and GDNF (Figure 7d). Importantly, the exacerbated neuropathic pain phenotype showed by cortistatin-deficient mice was reversed by chronic treatment with cortistatin (Figure 7a–d).

Similarly to the CCI model, induction of chronic diabetes in CST-KO mice resulted in more severe heat hyperalgesia and tactile allodynia (Figure 7e), as well as increased inflammatory markers in the sciatic nerve (Figure 7f), than those observed in CST-wt mice. These findings demonstrate that cortistatin could act as an endogenous break for developing neuropathic pain in response to peripheral nerve injury.

### 3.5. Cortistatin Promotes Functional Recovery after Sciatic Nerve Transection (SNT)

We next evaluated the interaction between peripheral nerve damage-associated neuropathic pain and nerve regeneration, and we asked whether cortistatin could influence neuropathic pain by affecting functional recovery and nerve regeneration and remyelination. After peripheral nerve transection in mice, denervated Schwann cells resume proliferation, which is followed by differentiation and remyelination of regenerated axons [51]. Thus, we subjected CST-wt and CST-KO mice to a model of severe nerve transection (Figure 8a), in which a segment of sciatic nerve was surgically removed and a tubular biodegradable conduit was used as a bridge that guides the growing axons between injured nerve stumps [52]. Sciatic nerve transection in mice causes impairment of the ankle extension, affecting the ability to increase the angle between the foot and leg (foot plantar flexion). Consequently, stepping with the injured leg changes from digitigrade (on the toes) to plantigrade (with the whole length of the foot on the ground). In addition, the toe spread on the injured side is reduced during walking compared with that of the unlesioned limb. The dysfunction of these parameters reflects both tibial and peroneal alterations as a consequence of sciatic nerve sectioning and led to gait disturbances (see Supplementary videos). Based on this, we selected and scored the gait movements, toe spreading, footsteps, and ankle flexibility as read-outs for the nerve functionality after sciatic nerve injury. We found that, despite the presence of the tubular guide, CST-wt animals reached a mild score (a score of 4 from a maximal value of 8) during the first week after surgery that slightly increased over time (Figure 8b). The lack of endogenous cortistatin exacerbated these locomotor dysfunctions, as CST-KO mice showed greater gait disturbances and reduced toe spreading, ankle and foot dorsiflexion, and lateral foot-base angle (LFBA) in comparison to CST-wt littermates (Figure 8b; Supplementary videos). Chronic treatment with cortistatin during six weeks around the transected sciatic nerve reduced the motor deficits and improved recovery caused by SNT in both CST-wt and CST-KO mice (Figure 8b). Interestingly, while cortistatin administration failed to significantly affect the ankle dorsiflexion impairment in lesioned CST-wt mice, it greatly influenced this parameter in cortistatin-deficient mice, reducing the early deficit in the external rotation of the foot and the impairment of plantar flexion, avoiding their worsening over time (Figure 8b), both consequences derived from an abnormal ankle angle during movement.

Because neuropathic pain is a well-known phenomenon in the sciatic nerve transection model, we next addressed the contribution of cortistatin to the correlation between motor dysfunction, nerve regeneration, and the recovery of sensory functions. For this, we measured the sensory neuropathic pain response to mechanical and thermal stimulation of the hind paws contralateral and ipsilateral to the transected nerve in the moment that nerve regeneration was in course. As expected, the ipsilateral paws showed increased allodynic and hyperalgesic responses in comparison to the contralateral side in CST-wt mice, and the deficiency in cortistatin significantly accentuated both nociceptive responses (Figure 8c). Importantly, the local injection of cortistatin during four weeks alleviated SNT-induced neuropathic sensitivity in CST-wt mice, and reversed the exacerbated phenotype observed in CST-KO mice (Figure 8c).

Together, these results suggest that cortistatin deficiency accelerates and accentuated nerve degeneration and retards functional recovery, while the exogenous administration of this neuropeptide positively influences motor and sensory functions after a complete severe lesion of the sciatic nerve.

### 3.6. Treatment with Cortistatin Attenuates Denervation-Induced Muscle Atrophy and Influences Nerve Regeneration after Sciatic Nerve Transection

Sciatic nerve transection leads to muscle atrophy, affecting muscle size, structure, contractile function, and mobility as a consequence of denervation [53]. In our STN model, denervation for six weeks caused the shrinkage of leg muscle and led to a reduction >60% in the mass of sciatic nerve-innervated gastrocnemius muscle in the lesioned leg compared with the contralateral side in both CST-wt and CST-KO (Figure 8d). Treatment with cortistatin of the lesioned nerve during six weeks significantly avoided the loss of muscle weight in both genotypes (Figure 8d). Histopathological analysis of the gastrocnemius muscle before damage showed a tightly packed homogenous polygonal shaped muscle fiber distribution with minimal intramyofiber spacing and uniform size. After nerve transection, a decrease in myofiber size (a reduction in the diameter of the muscle fiber) and an increase in the intramyofiber spacing can be observed. Accordingly, as the myofibers after injury were smaller, more fibers were quantified in the injured muscle compared with the same area in the uninjured muscle. This confirmed extensive muscle atrophy in the lesioned limb compared with the contralateral unlesioned leg in both CST-wt and CST-KO mice (Figure 8e). Treatment with cortistatin partially improved the derangement of the muscle fibers in the injured limbs of CST-wt mice, and significantly ameliorated the muscle atrophy observed in CST-KO mice by reducing the numbers of myofibers and the intramyofiber spacing (Figure 8e).

Finally, in order to examine the structural composition of injured nerves, we analyzed the presence of MBP+-myelinated axons in proximal and distal cross-sections of the sciatic nerve isolated six weeks after injury (Figure 8f). We found that, whereas the number of MBP+ axons was not affected by SNT at the proximal side, a significant reduction in myelinated nerve fibers occurred at the distal end of the lesioned nerve in comparison to the intact contralateral limb and to the proximal end of the same transected nerve (Figure 8f). Interestingly, this SNT-induced distal degeneration of myelinated axons was significantly more aggravated in CST-KO mice than in CST-wt mice (Figure 8f). Importantly, local treatment with cortistatin around the operated sciatic nerve increased almost twice the number of myelinated axons distal to the transection site in CST-wt mice and more than 3-folds in CST-KO animals (Figure 8f). These results support a key role for cortistatin in axonal recovery after nerve injury and a subsequent effect on recovering from denervated-induced muscle atrophy.

## 4. Discussion

Neuropathic pain is one of the most severe forms of chronic pain caused by the direct injury of the somatosensory system. The etiology factors that cause peripheral neuropathic pain are multiple and include injury of the peripheral nerves induced by compression and trauma, by exposure to various toxins and chemotherapeutic agents, by diabetic neuropathy, or by infection or autoimmune pathology [3,4]. The current analgesic drugs, including nonsteroidal anti-inflammatory drugs, opioids, antidepressants, and gabapentanoids, are inadequate to relieve chronic pain due to their limited efficacy or their significant side effects that limit their use [5,54]. Therefore, the development of analgesics that modulate new and/or multiple pharmacological targets and with novel modes of action is critical. In this study, we demonstrated that the treatment with the neuropeptide cortistatin alleviates chronic neuropathic pain in a variety of well-established preclinical models of chronic constriction and transection of the sciatic nerve and of diabetic peripheral neuropathy. Importantly, cortistatin injection alleviates two of major and severe manifestations of neuropathic pain such as mechanical and thermal allodynia and hyperalgesia. The demonstration of analgesic efficacy of cortistatin in different mouse models is of relevance from a therapeutic point of view, because, in general, there are differences between the pathophysiological mechanisms and responses to drug treatment for neuropathic pain induced by different types of injury to the peripheral nerves [54,55,56]. This means that cortistatin must tune down critical pathological mechanisms that are commonly involved in neuropathic pain of different etiologies.

Numerous clinical and preclinical studies have strongly demonstrated the importance of the complex interactions that exist between sensory neurons and nonneuronal cells in the generation and maintenance of neuropathic pain [45,46,47,48,49,50]. Thus, after peripheral nerve injury, many pathological changes occur, including the degeneration of nerve cells and axons and the dedifferentiation of Schwann cells, the release of inflammatory and cytotoxic mediators, the infiltration and activation of immune cells in the inflamed area, as well as microglia- and astrocyte-mediated neuroinflammatory processes at the peripheral and central nervous system levels [45,46,47,48,49,50,57]. In order to optimize pharmacological approaches to treat chronic neuropathic pain, it is necessary to regulate the nerve hypersensitization process from a perspective that includes both neuronal sensitization and neuroimmune interactions. Evidence indicates that cortistatin could regulate all these processes: neural sensitization, neuroimmune interactions, and nerve regeneration.

First, previous studies firmly demonstrated the capacity of cortistatin to regulate directly the process of the hypersensitization of primary nociceptive neurons by a variety of inflammatory and nociceptive stimuli [21,22]. This effect is independent of its anti-inflammatory activity, it is exerted on both peripheral and central-spinal nociceptive terminals, and it involves the impairment of nociceptive-induced intracellular signaling (protein kinase A, calcium, and Akt/ERK activation) and the expression and release of pronociceptive peptides and neurotransmitters [22]. Here, we confirmed these findings using a model of persistent inflammatory pain evoked with formalin, including the involvement of sstr and GHSR in the analgesic effects of cortistatin. As the sensitization of the primary nociceptive neuron at the peripheral and spinal levels also plays a crucial role in the generation and maintenance of neuropathic pain, and most of the nociceptive mediators (inflammatory cytokines, protons, prostaglandins, ATP, etc.) that activate nociceptors are shared by both inflammatory and neuropathic pain [45,47,48,49], one could assume that cortistatin would be able to tune down directly the neural activity through these same mechanisms. In fact, the injection of cortistatin rapidly alleviated allodynic and hyperalgesic responses in the different models of neuropathic pain used in this study, once the nerve injury was fully established, supporting a direct action on nociceptive sensitization. In both inflammatory and neuropathic pain, signaling though sstr and GHSR seems to be mainly involved in the primary analgesic effect of cortistatin ([21,22]; and this study), suggesting again that the molecular mechanisms are shared in both painful paradigms. Interestingly, we also found a partial involvement of opioid receptors at the central level during the initial neurogenic pain evoked by formalin and the neuropathic pain caused by nerve constriction, although no evidence to date has demonstrated the binding capacity of cortistatin to opioid-receptors. A factor such as cortistatin that acts through various anti-pain receptors in critical sites of the nociceptive system is highly attractive to develop analgesic therapies for neuropathic pain.

Second, evidence indicates that cortistatin could regulate the changes in the highly organized neuroimmune interactions initially provoked by neural damage. Thus, treatment with cortistatin reduced the inflammatory microenvironment that follows peripheral nerve injury. The decrease by cortistatin of the local release of the inflammatory cytokines TNFα, IL-6, and IL-1β, and other substances, such as prostaglandins and nitric oxide, which both sensitize the remaining intact axons and contribute to axonal damage [45,47,49,58], definitively contributes to initiate and maintain a long-lasting analgesic effect. Beside a direct anti-inflammatory effect on infiltrating immune cells and resident glial cells, cortistatin may also avoid the infiltration of macrophages and neutrophils to the inflamed nerve by decreasing the release of a chemokine, such as CCL2, that is critically involved in the progression of neuropathic pain [59]. These findings are supported by many studies reporting the anti-inflammatory action of cortistatin on macrophages, neutrophils, and microglia in other conditions through mechanisms that involve both sstr and GHSR and the regulation of multiple intracellular signals [15,27,60,61,62,63,64]. The effect of cortistatin on a wide panel of inflammatory mediators could suppose a clear advantage versus several therapies designed to neutralize single inflammatory cytokines/chemokines or to inhibit intracellular factors involved in inflammatory signaling (i.e., NFkB) that have proven their efficacy in models of neuropathic pain [48,65,66]. The anti-inflammatory actions of cortistatin in cells that are present on the damaged nerve could probably be extended to nonneural satellite cells of the corresponding DRGs and to glial cells of the spinal cord dorsal horn. In fact, we observed that treatment with cortistatin reduced the release of inflammatory cytokines and CCL2 in the spinal cord during chronic nerve injury, potentially reflecting a deactivation of microglia and astrocytes at the central level. This effect could have important pathological implications because the production of these inflammatory mediators facilitates the central transmission of pain signals in spinal nociceptors and is one of the hallmarks of neuropathic pain progression damage [47,49,50,57]. Thus, the regulation of inflammatory factors at the central level could contribute to the effect of cortistatin in modulating neuronal plasticity in secondary nociceptive neurons [22]. On the other hand, we found that, whereas the presence of cortistatin induced a less inflammatory peripheral nerve microenvironment, it increased or maintained the levels of two neurotrophins, such as GDNF and BDNF, which promote growth and regeneration of damaged axons [67,68]. As Schwann cells are the most abundant glial cells in the peripheral nervous system, are major producers of both neurotrophins, and acquire a repairing phenotype after a nerve is damaged [68], it is plausible to hypothesize that the effects of cortistatin in neurotrophins are exerted through these cells. However, the action of this neuropeptide on Schwann cells is fully unknown to date. Interestingly, in contrast to its effect on BDNF at the peripheral nerve level, cortistatin decreased the levels of this neurotrophin in the spinal cord after CCI. These apparent contradictory actions could be due, on one hand, to an inhibitory effect of cortistatin in the release by the central terminal of primary nociceptor at the spinal level of BDNF, thus preventing its capacity to sensitize sensory neurons and promote the development of neuropathic pain [69]; and, on the other hand, to a potential stimulatory effect on Schwann cells at the peripheral level that compensates a downregulation of axonal BDNF. Furthermore, some evidence indicates that BDNF could play pro-inflammatory actions, instead of neurotrophic effects, during neuropathic pain [70], a fact that might support the suppressive activity of cortistain in neuroinflammatory response in the injured nerve at the central and peripheral level. Moreover, we previously found that cortistatin induces the production of BDNF by immune cells [64], and therefore immune cells that are infiltrating the damaged nerve could contribute to compensate a potential loss of BDNF.

As previously described in several models of inflammatory pain [21,22], our study demonstrated that cortistatin acts as an endogenous natural analgesic that is produced in response to nociceptive stimuli, including nerve injury, in parallel to nociceptive peptides to mitigate persistent painful sensitization. We found that a deficiency in cortistatin exacerbated neuropathic pain and aggravated the disorganization of neuroimmune interactions discussed above, favoring an inflammatory milieu versus a nerve regenerative environment. Previous evidence demonstrated that cortistatin is produced by primary sensitive neurons that innervated laminae I–III of the spinal cord dorsal horn and by GABAergic interneurons localized in laminae IV–V and that exert inhibitory actions on the sensitization of spinal secondary nociceptors [22]. Endogenous production of cortistatin in both places could explain the analgesic effects proposed to regulate neuropathic pain at peripheral and central levels.

Third, besides the analgesic role of cortistatin, we demonstrated that cortistatin administration avoided the progression of motor deficits after sciatic nerve injury and recovered the normal pain threshold, and that these beneficial effects were accompanied by an enhancement of axonal regeneration and a partial reduction in the muscle atrophy. Moreover, the lack of cortistatin correlated with a significant aggravation of motor and sensory symptoms at early times, a failure in regeneration, a significant decrease in nerve remyelination, and an increased nociceptive response. Although nerves in the peripheral system can regenerate within a permissive environment, recovery is often incomplete and is characterized by poor axonal regrowth, muscle weakness, loss of sensation, and debilitating neuropathic pain, in many cases, refractory to analgesia [71]. Several approaches have been proven to exert beneficial effects on axonal repair, but none have achieved complete results, and a profound investigation on the reasons and therapies against poor recovery is needed [72]. Our data suggest that cortistatin could regulate nerve regeneration and functional recovery through direct and/or indirect effects on the sciatic nerve niche. In this, a cascade of inflammatory events, known as Wallerian degeneration, is triggered due to axonal degeneration distal to nerve damage, and ever precedes regeneration and nerve reconstruction [73]. This inflammatory environment is crucial for the dynamics of the Schwann cells, which first de-differentiate, proliferate, and adopt a repair phenotype, to subsequently differentiate and remyelinate the axons that finally reinnervate target organs [51]. Successful repair of the lesioned nerve requires coordination between cellular and molecular events integrated by Schwann cells and immune cells. Regarding immune events, although inflammation is necessary to reestablish tissue homeostasis, it can be harmful when exacerbated. As discussed above, excessive production of pro-inflammatory mediators and free radicals may cause damage to the nervous tissue, impairing axonal conductivity, causing retrograde neuronal death, and developing the conditions for neuropathic pain [6]. Regarding this, once cell fragments and myelin debris are cleared, the inflammatory response must be suppressed, creating a microenvironment prone to the neuroregenerative process [74]. Our data indicate that a deficiency in cortistatin accelerated and aggravated injury-induced sciatic nerve dysfunction and subsequently increased pain hypersensitivity, and that treatment with cortistatin, once the early cellular and molecular inflammatory events occurred, ameliorated both pathological processes without influencing the early inflammatory pre-reparative context. This is of relevance because when regeneration is advanced and the anti-inflammatory response is predominant [75], a second peak of hyperalgesia can be observed due to tissue remodeling and the appearance of newly regenerated sensory fibers [76]. These findings suggest a critical role played by cortistatin in the crosstalk between inflammation and repair during this period. In fact, as discussed above, treatment with cortistatin also promotes a neurotrophic and pro-regenerative environment at peripheral and central levels that is dysregulated when cortistatin is absent. Several neurotrophic factors have been used to promote nerve regeneration such as GDNF, artemin, NGF, and BDNF [77,78,79,80,81]. However, their therapeutic effects have been disappointing in most cases due to the short biological half-life and the insufficient concentrations at injury sites. Local periodic injections of these factors are impractical and expensive, and excessive doses may also evoke undesirable side effects. To solve the problems associated with using glial cell line-derived neurotrophic factors as drugs, it has been successfully proposed to target their receptors with small molecules for developing therapeutics for disease-modifying treatments against neuropathic pain [82,83,84]. Alternatively, a desirable therapy should be able to regulate the production of these mediators in a time- and concentration-dependent manner. As we demonstrated here, cortistatin modulates the production of these factors by DRGs, central terminals, and the sciatic nerve, and it restores the endogenous production of some of them (GDNF) while reducing others (BDNF) in an adequate balance at the lesion site (neurotrophic versus neuropathic actions), which could be determinant in the regenerative and neuroprotective effect of the neuropeptide in the repairing nerve. Schwann cells are key elements in the promotion of nerve regeneration. When a peripheral nerve is segmented, the proximal section still connected to the peripheral system remains intact, whereas that distal to the cut degenerates. Schwann cells from the cut ends may spread across the gap, forming a bridge and favoring the growth of the regenerating axons [51]. As stated above, although the effects of cortistatin on Schwann cell function are mostly unknown to date, we cannot discard a direct or indirect effect of cortistatin on Schwann cell dynamics. Thus, an initial transcriptomic study showed that Schwann cells increased the expression of sstr2 and cortistatin, but not of somatostatin or ghrelin, when subjected to a wound microenvironment and repairing program [85]. More recently, a multi-layered and single-cell transcriptome analysis of Schwann cells and the full sciatic nerve revealed that cortistatin and its receptors are expressed not only in Schwann cells but also in other cells of the sciatic nerve, including pericytes, epineurial, endothelial, and immune cells [86]. These findings suggest that Schwann cells and other accompanying cells that are crucial in supporting optimized peripheral nerve structure and function and in promoting axonal regeneration after nerve transection [87,88] are potential targets of cortistatin, which could act in an autocrine and/or paracrine manner in this context. Interestingly, the neuroprotective role of cortistatin was previously assessed in a model for multiple sclerosis in which treatment with cortistatin protected oligodendrocytes (the myelin-forming cells in the central nervous system) from oxidative stress [64]. In this sense, we recently demonstrated that oligodendrocytes that are deficient in cortistatin showed a genetic signature and phenotype compatible with a proliferative and immature profile and reduced myelinating capacity [89,90]. Moreover, the addition of cortistatin to co-cultures significantly increased the capacity of mature oligodendrocytes to re-myelinate axons of adult DRG neurons [89,90]. Despite their different location in the nervous system, the fact that they share a major function as myelinating cells probably supports that we could translate to Schwann cells the findings observed for cortistatin in oligodendrocytes, in order to explain their potential involvement in the phenotype observed in the transected nerve of cortistatin-deficient mice and the improvement found after the treatment with cortistatin.

In summary, it is evident that the nervous and immune systems interact and interfere in the success or failure of the regeneration process after neuronal injury [91]. Our data indicate that the modulation of the immune response is a key strategy for achieving better motor recovery following peripheral nerve injury. Cortistatin may promote repair, analgesia, axonal regeneration, and neuroprotection through both direct and indirect mechanisms that involve neuroimmune actions, the balance of neurotrophic factors, and/or the effects on the multicellular interactions in the sciatic nerve niche. On the other hand, endogenous production of cortistatin is critical in defining nociceptive and neuroregenerative responses and to understanding processes that occur during nerve repair in peripheral neuropathies.

Finally, our study has important clinical and therapeutic implications, both related to the design of new multitargeted therapeutic strategies based on the use of cortistatin for treating chronic neuropathic pain of different etiologies, and to the identification of cortistatin as a potential risk factor for developing severe neuropathic pain. It is obvious that a therapeutic agent such as cortistatin with the capacity to regulate several of the critical pathological mechanisms involved in the generation and maintenance of neuropathic pain is an advantage against other approaches or current drugs. It is important to mention that cortistatin has a favorable safety profile in humans and demonstrated clinical efficacy in patients with Cushing’s disease [92], supporting its further bedside translation. Although one of the potential limitations of translating cortistatin-based therapies to the clinical practice is its peptidic nature and low stability in physiological fluids, the interest of the pharmaceutical industry in protecting it and improving its bioavailability has considerably increased lately. For example, the design of a new cortistatin-based analog with improved half-life in serum and therapeutic efficacy in inflammatory conditions has recently consolidated this strategy [93]. On the other hand, the identification of cortistatin as an endogenous break for nociception could help to explain the differential painful responses, in terms of intensity and duration, that are observed in many patients with a particular neuropathy. For example, about 20–60% of patients with diabetes mellitus or diabetic neuropathy develop neuropathic pain, and the reason why some patients experience it while others do not is not fully understood. As some risk factors, including demographic, metabolic, sensory, and genetic factors, have been suggested to contribute for developing diabetic neuropathic pain [94], it is intriguing to speculate the possibility that a deficiency in cortistatin could predispose these patients to suffering diabetic pain. If this would be the case, our findings also demonstrated that this deficiency and susceptibility could be easily corrected by a cortistatin-based treatment.

## Figures and Tables

**Figure 1 pharmaceutics-13-00947-f001:**
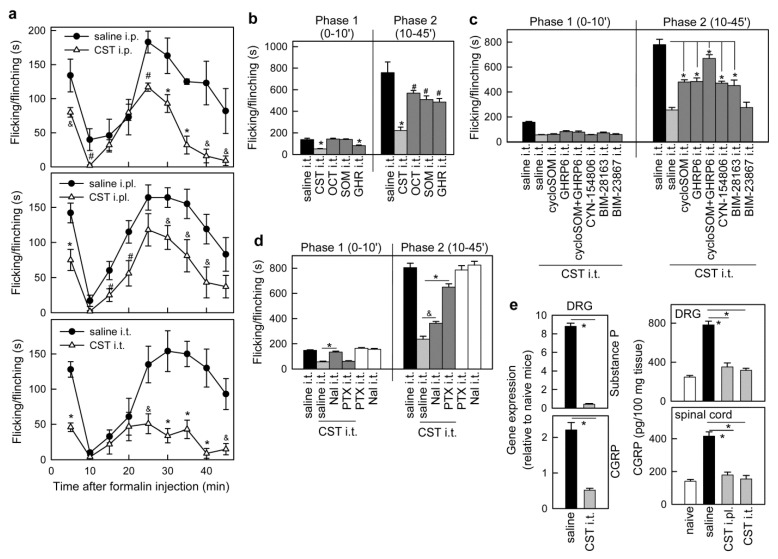
Peripheral and central analgesic actions of cortistatin on formalin-evoked acute pain. (**a**) Effects of central (i.t., 10 ng), peripheral (i.pl., 100 ng), or systemic (i.p., 1 µg) cortistatin (CST) injection in formalin-induced acute inflammatory pain responses. *n* = 6–8/group. * *p* < 0.0001, & *p* < 0.01, # *p* < 0.05 vs. saline-treated mice. (**b**) Comparative effects of i.t. administration of cortistatin, somatostatin (SOM), octreotide (OCT), and ghrelin (GHR) (10 ng each) in formalin-induced nocifensive responses. *n* = 6–7/group. * *p* < 0.0001, # *p* < 0.05 vs. saline-treated mice. (**c**) Reversal of cortistatin-mediated attenuation of formalin-induced pain behavior by antagonists against all sstr (cycloSOM), ghrelin-receptor (GHRP6 and BIM-28163), sstr2 (CYN-154806), and sstr5 (BIM-23867) administered centrally (i.t.). *n* = 7/group. * *p* < 0.0001 vs. cortistatin-treated mice. (**d**) Reversal of cortistatin-mediated attenuation of formalin-induced pain behavior by opioid-receptor (naloxone, Nal) and by pertussis toxin (PTX) administered centrally (i.t.). *n* = 6–7/group. * *p* < 0.0001, & *p* < 0.01 vs. cortistatin-treated mice. (**e**). Central (i.t., 10 ng) and peripheral (i.pl., 100 ng) actions of cortistatin in the expression of Substance P and calcitonin gene-related peptide (CGRP) in formalin-evoked pain determined by RT-PCR (left panel, 30 min after cortistatin injection) or by ELISA (right panel, 1 h after cortistatin injection) in ipsilateral lumbar dorsal root ganglia (DRG) and dorsal horn spinal cords. Naïve animals were used as basal controls of reference. *n* = 4/group. * *p* < 0.0001 vs. saline-treated mice. All data represent mean ± SEM. i.t., intrathecal; i.p., intraperitoneal; i.pl., intraplantar; sstr: somatostatin-receptors.

**Figure 2 pharmaceutics-13-00947-f002:**
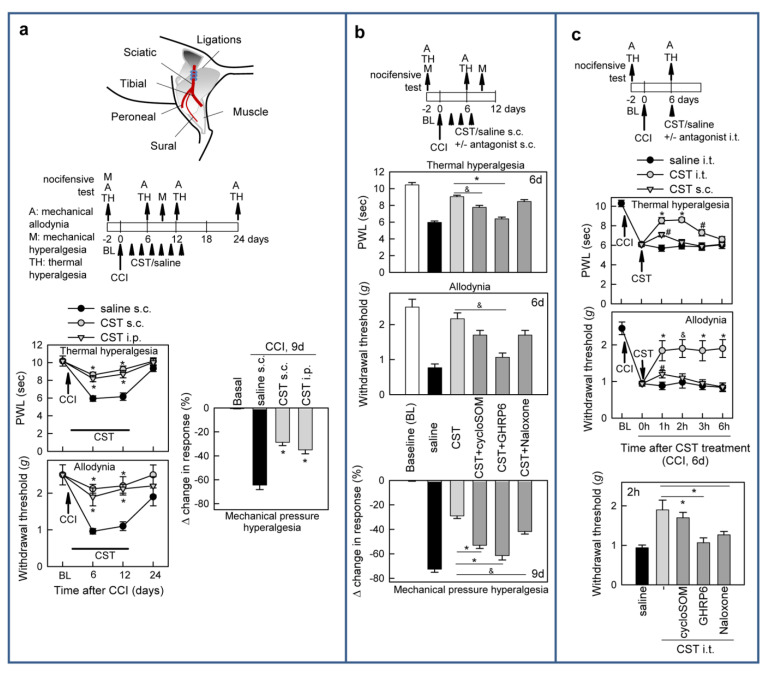
Cortistatin attenuated neuropathic pain caused by chronic constriction injury (CCI) of the sciatic nerve. (**a**) Effects of local (s.c. 1 µg) and systemic (i.p., 2 µg) cortistatin injection (every other day starting 2 days after CCI surgery until day 13, as indicated in the scheme) on thermal hyperalgesia represented by paw withdrawal latency (PWL) in response to radiant heat, on mechanical/tactile allodynia represented by the withdrawal threshold (in g) to plantar von Frey hair application in hind paws (ipsilateral to injured nerve), and on mechanical pressure hypersensitivity represented as the percentage of decrease in the pressure threshold in the ipsilateral paw over the contralateral paw to injured nerve, prior to CCI (BL, baseline) or at different times after CCI surgery. *n* = 6 mice/group. * *p* < 0.0001 vs. saline-treated mice. (**b**) Reversal of cortistatin-induced relief of CCI-induced pain behavior by antagonists of sstr (cycloSOM), ghrelin-receptor (GHRP6), and opioid-receptor (naloxone) administered locally (s.c.), as described in the scheme. *n* = 6 mice/group. * *p* < 0.0001, & *p* < 0.01 vs. cortistatin-treated mice. (**c**) Effects of a single local (s.c. 1 µg) or central (i.t., 20 ng) cortistatin injection on thermal hyperalgesic and allodynic responses measured 6 days after CCI induction. Lower panel: effects of i.t. administration of antagonists for sstr, ghrelin-receptor, and opioid-receptor on the anti-allodynic effect of cortistatin (measured 2 h after i.t. cortistatin injection). *n* = 6 mice/group. * *p* < 0.0001, & *p* < 0.01, # *p* < 0.05 vs. saline-treated mice in upper panels, and vs. cortistatin-treated mice in lower panel. All data represent mean ± SEM.

**Figure 3 pharmaceutics-13-00947-f003:**
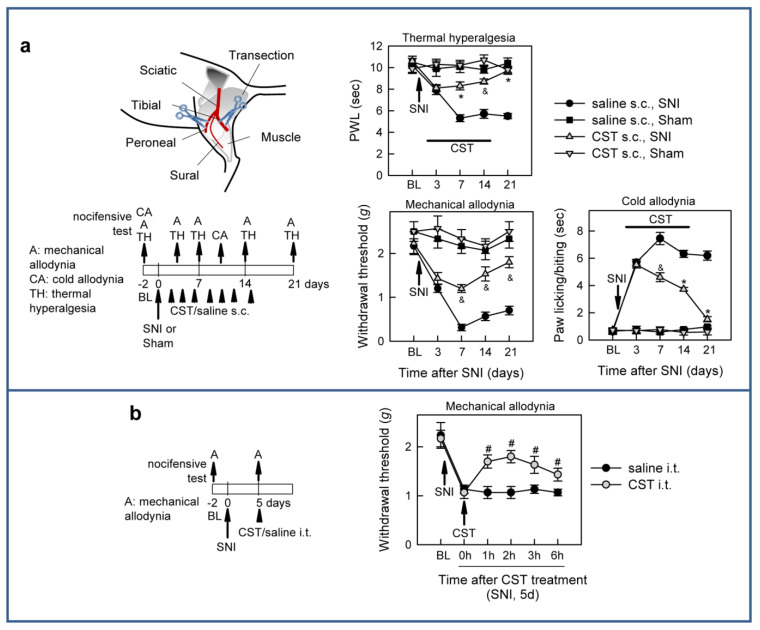
Cortistatin alleviated thermal hyperalgesia and allodynia in a model of spared nerve injury (SNI). (**a**) Effects of local (s.c. 1 µg) cortistatin injection (every other day starting 2 days after SNI-surgery until day 15, as indicated in the scheme) on thermal hyperalgesia represented by paw withdrawal latency (PWL) in response to radiant heat, on mechanical allodynia represented by withdrawal threshold (in g) to plantar von Frey hair application in hind paws ipsilateral to the surgery, and on cold allodynia determined by duration of hind paw licking and biting in response to acetone application to the paw ipsilateral to the surgery, before (BL, baseline) or at different times after SNI. Sham-operated mice were used as controls of reference. *n* = 6 mice/group. * *p* < 0.0001, & *p* < 0.01 vs. saline-treated mice. (**b**) Effects of a single central (i.t., 20 ng) cortistatin injection on mechanical allodynia measured 5 days after SNI induction. *n* = 6 mice/group. # *p* < 0.05 vs. saline-treated mice. All data represents mean ± SEM.

**Figure 4 pharmaceutics-13-00947-f004:**
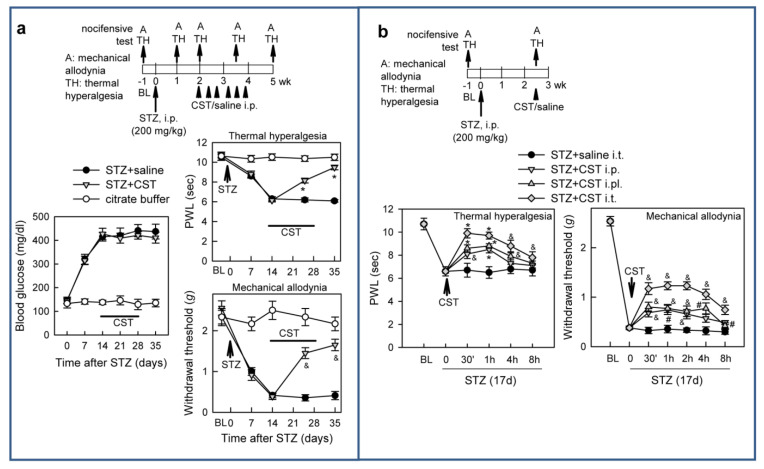
Cortistatin reduced neuropathic pain in diabetic mice. (**a**) Effects of systemic (i.p. 2 µg) cortistatin injection (every other day, for two weeks, as indicated in the scheme) on thermal hyperalgesia represented by paw withdrawal latency (PWL) in response to radiant heat, and on mechanical allodynia represented by the withdrawal threshold (in g) to plantar von Frey hair application in hind paws, before (BL, baseline) or at different times after induction of chronic diabetes by a high dose of streptozotocin (STZ). Nondiabetic mice injected with citrate buffer instead of STZ were used as controls of reference. The course of diabetes was monitored by measuring the glucose levels in blood (left panel). *n* = 6–8 mice/group. * *p* < 0.0001, & *p* < 0.01 vs. saline-treated diabetic mice. (**b**) Effects of a single injection of cortistatin at local (i.pl., 1 µg), systemic (i.p., 2 µg), or central (i.t., 20 ng) levels on thermal hyperalgesia and mechanical allodynia measured 17 days after STZ infusion. *n* = 8 mice/group. * *p* < 0.0001, & *p* < 0.01, # *p* < 0.05 vs. saline-treated STZ-diabetic mice. All data represents mean ± SEM.

**Figure 5 pharmaceutics-13-00947-f005:**
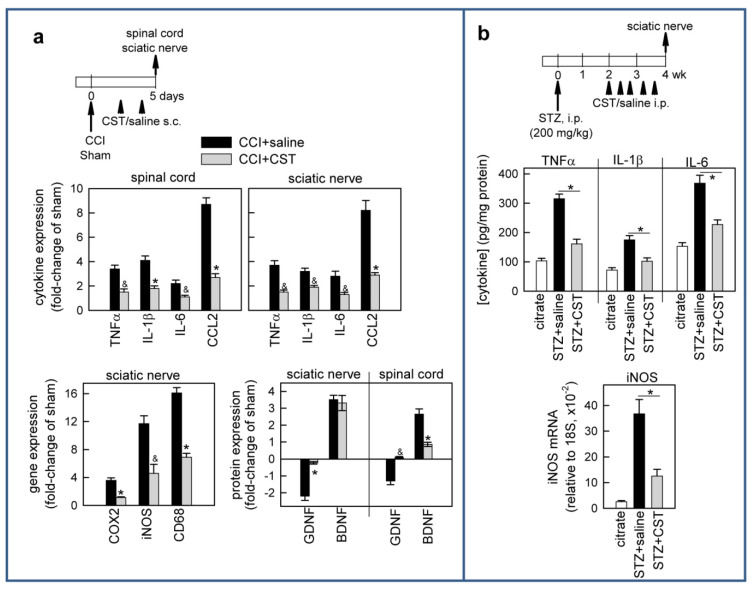
Treatment with cortistatin regulated neuroinflammation and neurotrophic factors in nocifensive system after peripheral nerve injury. (**a**) Effects of cortistatin (1 μg, injected s.c. around the injured nerve, as indicated in scheme) on the contents of cytokines, chemokines, and neurotrophic factors in protein extracts and on gene expression of COX2 and iNOS in mRNA isolated from ipsilateral sciatic nerve and L4-L5 dorsal spinal cord segments dissected at day 5. The presence of macrophages in the sciatic nerve was determined by measuring the expression of CD68. Results were referenced to samples obtained from sham-operated mice. *n* = 6 mice/group, pooled in three samples. * *p* < 0.0001 versus saline-treated CCI mice. (**b**) Effects of cortistatin (2 μg, injected i.p., as indicated in scheme) on cytokine contents in protein extracts and on iNOS gene expression in mRNA isolated from sciatic nerves four weeks after injection of streptozotocin (STZ). Nondiabetic mice (citrate) were used as controls of reference. *n* = 8 mice/group. * *p* < 0.0001 versus saline-treated STZ-diabetic mice. All data represent mean ± SEM.

**Figure 6 pharmaceutics-13-00947-f006:**
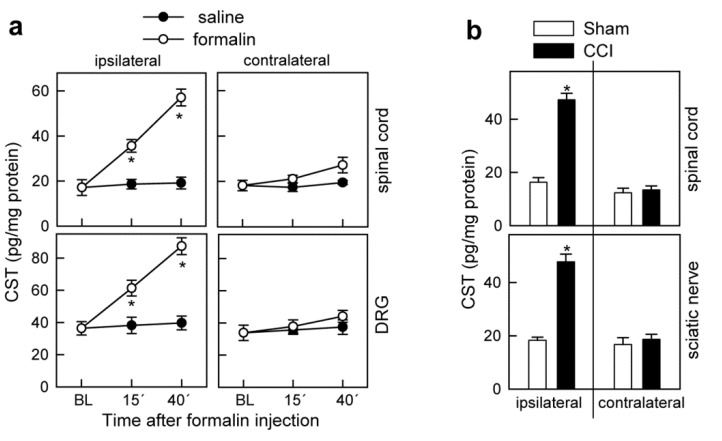
Production of cortistatin by the nociceptive system during inflammatory and neuropathic pain. (**a**) Cortistatin levels in lumbar dorsal root ganglia (DRG) and dorsal spinal cord (ipsilateral and contralateral to the injected paw) isolated before (BL, baseline) or at different times after i.pl. injection of formalin. Mice i.pl. injected with saline were used as a reference. *n* = 5 mice/group. (**b**) Cortistatin levels in the sciatic nerve and spinal cord ipsilateral and contralateral to surgery, measured five days after CCI. Sham-operated mice were used as a reference. *n* = 6 mice/group. * *p* < 0.0001 versus saline-treated mice (in a) or sham-operated mice (in b). All data represent mean ± SEM.

**Figure 7 pharmaceutics-13-00947-f007:**
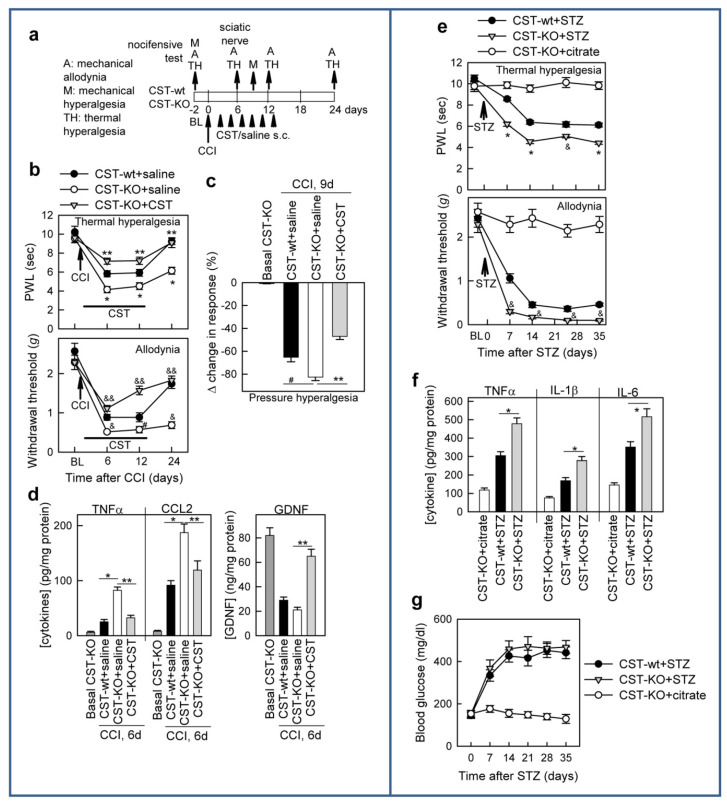
Deficiency in cortistatin exacerbated neuropathic pain in various models of peripheral nerve injury. (**a**–**d**) Effects of cortistatin deficiency (CST-KO mice) on the development of thermal hyperalgesia (**b**, upper panel), mechanical/tactile allodynia (**b**, lower panel), and mechanical pressure hypersensitivity (**c**) in hind paws (ipsilateral to injured nerve) after CCI of sciatic nerve in comparison to wild-type mice (CST-wt). Reversion of the exacerbated nocifensive responses to CCI of sciatic nerve observed in CST-KO mice by local s.c. cortistatin treatment was evaluated, as indicated in the scheme (**a**). The levels of inflammatory cytokines and GDNF in protein extracts of sciatic nerves isolated six days after CCI-surgery are shown (**d**). *n* = 6–7 mice/group. * *p* < 0.0001, & *p* < 0.01, # *p* < 0.05 between CST-KO and CST-wt mice. ** *p* < 0.0001, && *p* < 0.01, between saline-treated and CST-treated CST-KO mice. (**e**–**g**) Effects of cortistatin deficiency (CST-KO mice) on the development of thermal hyperalgesia (**e**, upper panel) and mechanical allodynia (**e**, lower panel) in hind paws during the course of diabetes induced by high-dose of streptozotocin (STZ) in comparison to wild-type mice (CST-wt). The levels of inflammatory cytokines in protein extracts isolated from sciatic nerves 28 days after STZ injection are shown (**f**). Nondiabetic CST-KO mice injected with citrate buffer instead of STZ were used as controls of reference. The course of diabetes was monitored by measuring the glucose levels in blood (**g**). *n* = 6–7 mice/group. * *p* < 0.0001, & *p* < 0.01 between CST-KO and CST-wt STZ-diabetic mice. All data represent mean ± SEM.

**Figure 8 pharmaceutics-13-00947-f008:**
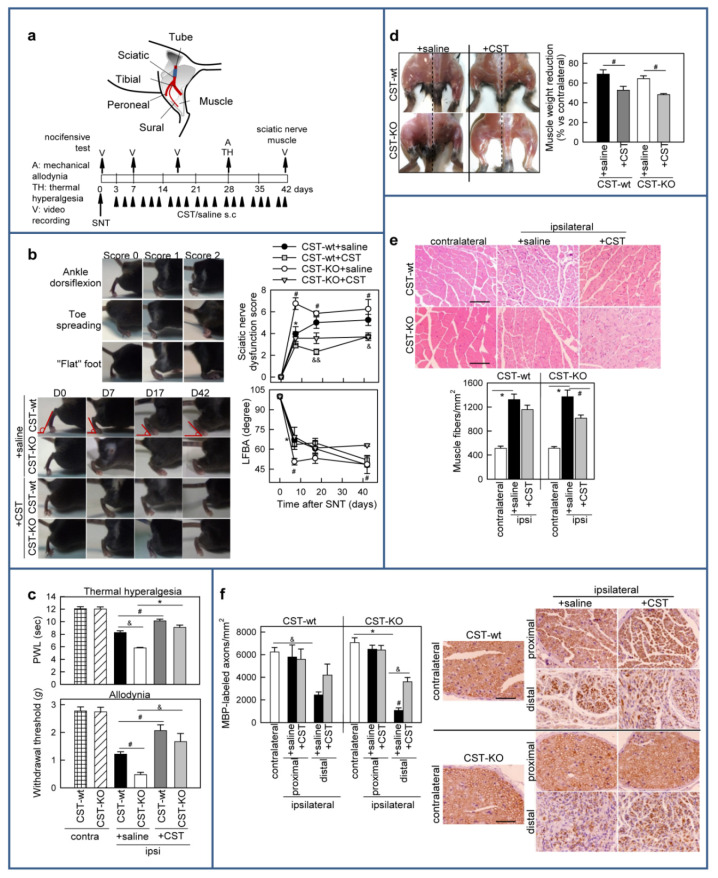
Cortistatin enhances functional recovery and sciatic nerve remyelination after severe nerve transection. (**a**) Schematic representation for the experimental design in the sciatic nerve transection (SNT) model. (**b**) Effects of cortistatin deficiency (CST-KO vs. CST-wt) and of local (s.c. 3 µg) cortistatin administration (every other day, for six weeks, as indicated in the scheme; CST-wt + CST, CST-KO + CST) on the development and progression of sciatic nerve dysfunction after severe SNT. Upper panel: the sciatic nerve dysfunction was quantified as the total sum of the scores observed for motor activity integrated by gait abnormalities (see Supplementary videos), ankle dorsiflexion, toe spreading, and “flat” foot presence (each parameter was individually scored from 0 to 2, as shown in the representative video frames). Lower panel: lateral foot-base angle (LFBA) was measured using 4–6 side-view frames for the ipsilateral foot and quantifying the angle formed by the horizontal line parallel to the floor and the line parallel to the mid- and hindfoot (excluding the toes; red lines). *n* = 4–7 mice/group. * *p* < 0.05 between saline-treated CST-wt and CST-KO; # *p* < 0.05 between saline-treated and CST-treated CST-KO mice; & *p* < 0.05, && *p* < 0.01 between CST-treated and saline-treated CST-wt. (**c**) Effects of cortistatin deficiency and treatment in SNT-induced thermal hyperalgesia and tactile allodynia determined in the hind paws of the lesioned (ipsi) and unlesioned (contra) side 28 days post-surgery. *n* = 4–7 mice/group. # *p* < 0.05, & *p* < 0.01, * *p* < 0.0001. (**d**) Effects of cortistatin deficiency and treatment in SNT-induced muscle atrophy. Representative photographs of tibialis anterior muscles isolated from saline-treated and cortistatin-treated lesioned legs of CST-wt and CST-KO mice are shown, and the wet weight for the gastrocnemius muscle was calculated for operated (IL) vs. contralateral (CL) hind leg 42 days post-surgery. *n* = 4–7 mice/group. # *p* < 0.05 vs. saline-treated mice in each genotype. (**e**) Effects of cortistatin deficiency and treatment in gastrocnemius muscle myofiber morphology. Representative images of hematoxylin/eosin-stained cross-sections of contralateral and ipsilateral muscles of saline- and cortistatin-treated CST-wt and CST-KO mice are shown and were used for quantifying the density of myofibers. *n* = 4–7 mice/group. Scale bars, 100 µm. * *p* < 0.0001 vs. unlesioned contralateral muscle for each genotype; # *p* < 0.05 vs. cortistatin-treated CST-KO mice. (**f**) Effects of cortistatin deficiency and treatment in axonal demyelination after SNT. The density of MBP-labeled axons within the proximal and distal nerve ends of the transected nerve was evaluated in cross-sections (representative images of MBP-immunostaining are shown; scale bars: 50 µm) of sciatic nerves isolated from saline- and cortistatin-treated CST-wt and CST-KO mice. Sections of contralateral intact nerves obtained in the same locations than of ipsilateral nerves were used as controls of reference. *n* = 4–7 mice/group. & *p* < 0.01, * *p* < 0.0001 vs. unlesioned contralateral; & *p* < 0.01 vs. lesioned saline-treated CST-KO; # *p* < 0.05 between lesioned saline-treated CST-KO and CST-wt. All data represent mean ± SEM.

## Data Availability

The data that support the findings of this study are available from the corresponding author upon reasonable request.

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
