# Peer review of "The Neuropeptide Cortistatin Alleviates Neuropathic Pain in Experimental Models of Peripheral Nerve Injury"

_pharmaceutics, 2021, doi:10.3390/pharmaceutics13070947_

Round 1

Reviewer 1 Report

This manuscript describes the analgesic efficacy of the neuropeptide cortistatin in different models of pain, including inflammatory and neuropathic pain.

The experiments are well designed and performed and clearly demonstrate that cortistatin not only is effective by acting at several levels of the pain neuraxix but also is an important modulator of the neuroimmune response in neuropathic conditions. 

Authors have demonstrated that cortistatin is effective after local, systemic or central administration and that the analgesic effect combines several mechanisms including a direct action on nociceptors, antinflammatory properties and the ability of modulating levels of neurotrophic factors involved in the generation and maintenance of chronic pain.

The paper is well written and results clearly described.

Author Response

We really appreciate the positive comments of this reviewer.

Reviewer 2 Report

The current work did comprehensive work on the effects of cortistatin in analgesia of different experimental models. The study also suggested the possible mechanisms and the therapeutic roles of cortistatin in pain relief. Overall, the results supported authors' hypothesis. Some minor revisions might be made:

  1. The abbreviations like BDNF and GDNF please give the full name when first mentioned;
  2. The image quality of Figure 8d can be improved;
  3. What is the gender of mice used in this study? Although the author mentioned there is no sex difference. But the estrogen levels in females could affect the  interpretation of data;
  4. The authors showed in Figure 6 that the production of CST was drastically increased in both inflammatory and neuropathic pain. While additive cortistatin is able to attenuate pain. Does it suggest the sensitivity to cortistatin is changed in those models?

Author Response

We thank the positive and constructive suggestions of this reviewer, and that he/she has only found minor concerns.

  1. We have followed the suggestion of this reviewer of giving the full name of GDNF and BDNF when first mentioned (page 8, paragraph 2.11).
  2. We have improved the quality of the figure 8. All original figures will be uploaded with high resolution as indicated by the journal instructions.
  3. We used a mix of male and female mice of the same age and in most of the cases of the same litter. We fully agree with this reviewer that estrogens and other sexual hormones, as well as the potential differential behaviours between genders could affect the nociceptive responses, and therefore, it is critical for us to use both males and females in every study in order to confirm that results do not depend on the gender and data can be extrapolanted to both sexes. The use of both males and females in the study was included in methods (page 3, paragraph 2.1).
  4. Our hypothesis is that the increase in the levels of cortistatin during inflammatory and neuropathic pain is a natural analgesic response of the nociceptive system to compensante or balance the production of proalgesic mediadors and peptides like CGRP, substance P, glutamate. This response is not unique for cortistatin, because other analgesic peptides and neurotransmitters, like somatostatin, GABA, neuropeptide Y, opioids,..., are secreted in response to pain induction. Exogenous addition of cortistatin, or any of these other natural analgesic factors, in general at higher doses than that endogenously produced, overcompensates the proalgesic responses and helps to the endogenous cortistatin. A critical set of data that confirm this hypothesis is obtained with cortistatin-deficient mice, in which in the absence of endogenous cortistatin, the nociceptive response is exacerbated in all the models assayed. This issue was discussed in page 23, last paragraph.

Reviewer 3 Report

Enjoyed reading this original article, very deep knowledge of the topic, strong discussion part. There are several questions for the authors.

  1. Why did you use the glucometer instead of other methods of measuring blood glucose?
    What time was the analysis carried out, how many hours of fasting was there before the analysis? This should be mentioned in materials and methods.
  2. Authors mentioned about regeneration capacity of neurotrophic factors such as GDNF, NGF and BDNF and that their therapeutic effects have been disappointing in most cases due to the short
    biological half-life and the insufficient concentrations at injury sites. In this case, authors can cite original articles about small molecules with the same mechanism of action such as NTFs. For example, glial cell line-derived neurotrophic factors (GFLs) proteins are not drug-like; they have poor pharmacokinetic properties and activate multiple receptors. Targeting RET and/or GFRα with small molecules may resolve the problems associated with using GFLs as drugs and can result in the development of therapeutics for disease-modifying treatments against neuropathic pain.
    Viisanen H, Nuotio U, Kambur O, Mahato AK, Jokinen V, Lilius T, Li W, Santos HA, Karelson M, Rauhala P, Kalso E, Sidorova YA. Novel RET agonist for the treatment of experimental neuropathies. Mol Pain. 2020 Jan-Dec;16:1744806920950866. doi: 10.1177/1744806920950866. PMID: 32811276; PMCID: PMC7440726; Sidorova YA, Bespalov MM, Wong AW, Kambur O, Jokinen V, Lilius TO, Suleymanova I, Karelson G, Rauhala PV, Karelson M, Osborne PB, Keast JR, Kalso EA, Saarma M. A Novel Small Molecule GDNF Receptor RET Agonist, BT13, Promotes Neurite Growth from Sensory Neurons in Vitro and Attenuates Experimental Neuropathy in the Rat. Front Pharmacol. 2017 Jun 21;8:365. doi: 10.3389/fphar.2017.00365. PMID: 28680400; PMCID: PMC5478727; Mahato, A.K., Sidorova, Y.A. Glial cell line-derived neurotrophic factors (GFLs) and small molecules targeting RET receptor for the treatment of pain and Parkinson’s disease. Cell Tissue Res 382, 147–160 (2020). https://doi.org/10.1007/s00441-020-03227-4
  3. Why is such a promising molecule still not used in clinical practice? Are there clinical trials? Are there ways to improve the pharmacokinetic properties of cortistatin?
  4. Line 599 two dots instead of one.

Author Response

We appreciate the positive words of this reviewer and we are happy to hear that he/she has enyoied reading our manuscript. Here are the responses to the well-suggested minor concerns:

  1. We usually use glucometer and electronic sticks form Bayer to measure glucose levels in an small (10 ul) quantity of blood obtained from the tail vein. This is a conventional, very reproducible and painless methodology that does not significantly affect the integrity and behaviour of the animal. Measurement was done at the same moment (at 10:00 a.m.) everytime. Animals were not subjected to fasting before glucose measurement, because one established diabetes two weeks after STZ injection, hyperglicemia is chronically maintained for more than 5 weeks. As suggested this reviewer, we have included this information in the methods section (page 6, paragraph 2.7).
  2. We appreciate this excellent suggestion of this reviewer of discussing the alternative of targeting RET and/or GFRα with small molecules in order to resolve the problems associated with using glial cell line-derived neurotrophic factors. The discussion about this issue is now included in page 25 (first paragraph), and the suggested references have been included in the reference list.
  3. One of the problem of translating results with neuropeptides and hormones to clinical practice is the low stability of these molecules in solutions, mainly in serum. Despite this potential disadvantage, the naive peptide is highly efficient in vivo. In fact, cortistatin has been previously used in patients with Cushing's disease, showing safety and efficiency. However, most of the companies are trying to discover small molecules that mimic cortistatin-mediated signalling, something that results quite difficult to reach, because cortistatin exerts most of its effects through various non-related receptors (i.e., sstr and GHSR). Another strategy from the pharmaceutical industry is increasing the stability of cortistatin in serum, designing new analogues by aminoacid substitutions/modifications of the naive structure of cortistatin. An example has been recently reported by our group in Nat. Commun. (Rol et al., 2021), in which an stable analogue of cortistatin showed similar efficiency than cortistatin ameliorating experimental inflammatory intestinal disease and even better than conventional treatments currently used in the clinic for the treatment of these patients. In this sense, this field is significantly progressing to the clinical translation in many disorders, and chronic pain could be one of the next to be addressed, and therefore, this study is covenient to communicate at this moment. This issue is addressed at the end of discussion (page 26).
  4. This mistake of edition was adequetely corrected.     

Reviewer 4 Report

Neuropathic pain results from nerve injury led by various causes and its prevalence has been increasing worldwide. Because alleviating neuropathic pain still remains an unmet medical need despite numerous efforts being made to treat it, alternative approaches are required. In this manuscript, Falo et al focused on a cyclic neuropeptide cortistatin known as a potent anti-inflammatory molecule to induce relief of inflammatory pain and demonstrated its analgesic activity in neuropathic pain. This study prove cortistatin is an attractive seed in drug development for the treatment of neuropathic pain and opened up the cortistatin-based therapy.

The study is thoroughly performed and the data are convincing. This work is of interest and value of readers of Pharmaceutics. Therefore this manuscript is strongly recommended for publication in the journal, as it is.

Author Response

We really appreciate the positive words of this reviewer about our study.